# The Effect of Leisure Time Sport on Executive Functions in Danish 1st Grade Children

**DOI:** 10.3390/children9101458

**Published:** 2022-09-23

**Authors:** Gitte Meilandt Siersbaek, Mona Have, Niels Wedderkopp

**Affiliations:** 1Bornespecialisterne, Bronzevej 11, 5250 Odense, Denmark; 2Danish School of Education, Campus Emdrup, 2400 Copenhagen, Denmark; 3Department of Clinical Research, University of Southern Denmark, 5230 Odense, Denmark

**Keywords:** leisure time sport, executive functions, children, inactivity, physical activity

## Abstract

Physical inactivity can influence children’s executive functions with severe impact on wellbeing and academic learning. The objective is to study the effect of leisure time sport on executive functions in Danish 1st grade children, and secondary to explore if socio-economy is a confounder for associations between leisure time sport and executive functions. This study is a sub-study nested within a cluster-randomized controlled trial with two arms (ClinicalTrials.gov, NCT02488460). 505 children from twelve schools, mean age 7.2 ± 0.3 years participated. Outcomes for executive function were “Modified Eriksen Flanker/Reverse Flanker Task” and “Behavior Rating Inventory of Executive Function” (BRIEF-P). Parents used SMS-tracking to register their children’s leisure time sport. Multivariate analyzes was performed using mixed linear regression, with adjustment for highest parental education, sex, municipality, and school-type. We found that leisure time sport seems to significantly improve working memory (WM) with nearly 20%, and furthermore it seems to be a significant predictor of ‘Initiate’ (the ability to begin an activity, to generate ideas, responses or problem-solving strategies). Socio-economy was not found to be a confounder. This study lends support to the hypothesis that leisure time sport is related to working memory capacity in children.

## 1. Introduction

Several aspects of children’s health and wellbeing are depending on familial and social factors, together with the constructs of society [1]. Building healthy activity behaviors, such as participating in leisure time sport (LTS) early in life is a central strategy in childhood health and wellbeing in Denmark [2]. Parents and other primary caregivers are the key influencers in the development of the behaviors through their parenting practices [2,3,4].

Nevertheless, parents’ time for participating in LTS, together with one’s children is lacking [1,5]. Unfortunately, time pressure is an escalating problem probably due to economic, cultural, and technological changes of society [5].

Also, numerous children and adolescents spend a considerable amount of time on electric devices [6,7]. This behavior might influence their sleep pattern [8,9,10,11], and the routines of everyday life such as the motivation for doing other things (like exercising). Sleep deprivation and lack of exercise for a longer period, can affect wellbeing, health and behavior, hence the risks of psychological distress increases [7,12,13,14]. Psychological distress could be dangerous for several parts of the brain, including the very important executive functions (EF), which might suffer to a great extent [11,13,15,16].

### 1.1. Executive Functions

EF refers to the ability to solve problems from higher perspectives, to change strategies during activities, or to understand what another person could be feeling or why they are acting in a certain way [14]. EF develops with age during childhood and early adulthood [16,17]. Cognitive associations with aerobic fitness in young adults and children have been published [18,19,20,21,22,23,24], especially in tests assessing executive functions (EF) which constitutes an important predictor for academic achievement [25,26,27,28,29,30,31]. This association has led to the EF hypothesis that physical activity could stimulate neural development and synaptic transmission and thus improve thinking, decision-making, and behavior in prefrontal cortices linked to executive functions [32].

Although still a matter of debate [33], one prominent theoretical framework suggests that executive function consists of three foundational components: inhibition, working memory, and cognitive flexibility [16,34]. Inhibition involves resisting temptation and being able to regulate one’s attention, behavior, thoughts, and emotions to override a strong internal predisposition or external lure for impulsive action. Working memory is the ability to temporarily hold and manipulate information and is closely related to problem-solving using previously learned information. Cognitive flexibility is the ability to switch perspectives and focus of attention and to adjust to changed demands and priorities [16,34].

Cognitive flexibility requires and builds on inhibitory control and WM, thus one need to inhibit a previous perspective and load into WM (or activate) a different perspective, to change perspectives [16].

### 1.2. Leisure Time Sport and Executive Functions

There is substantive evidence of children participating in sport gain psychological and social health benefits [34]. And children who are not exercising as a natural part of their life, would not integrate exercising as a habit into adulthood, and thereby they increase the risk of lifestyle diseases, obesity and low self-esteem when growing up [35].

Previous studies have found positive correlations between LTS participation and EF [14], and LTS and cognition [36,37].

In the Danish society both parents often work away from home [1] and the children thereby spent 8–9 h in kindergarten, school or engaging in afterschool social programs, Monday to Friday. Only the evenings and weekends are open for family activities, but this is also the time for the parents to practice their leisure time activities, to meet up with friends, clean the house and so on, not rarely leaving the children on their own, doing whatever they like. Thus, a paradox arises between the lack of time and motivation for children and their parents to exercise, and the risk of not acquiring the positive development of EF that exercise seems to create.

When new things are learned, the brain develops new connections [38,39,40]. This ability to shape new neurons, called neurogenesis, and the brains ability to strengthen “old” pathways, called long-term potentiation, is of great importance when learning new skills. Physical activity (PA) has been shown to support these abilities in the hippocampus [38,41,42,43], and in the prefrontal cortex [37,44], an important brain area when we discuss EF [15,45].

Research aimed at the relationship between PA and EF in children is still in its early stages. Only a few studies, in human, of more rigorous design with few bias or systematic errors, e.g., randomized controlled trials (RCT), are conducted [41]. Nevertheless, in three of those RCT’s it has been concluded that PA has a causal effect on cognitive performance in children [37,44,46].

The effect on EF differs depending on the “time-window”. It could be the effect from PA on EF right after a single bout of exercise, denoted as “acute PA” [47,48,49], or the effect from PA on EF after repetitive bouts of activity over longer periods of time, denoted as “chronic PA” [47,48,49]. Chronic PA is actual for this present study. De Greeff and colleagues conducted a meta-analysis, with a total of 31 included studies and 4593 children [48], demonstrating no acute effects from a single bout of exercise on EF. However, effects on EF were found from chronic PA, and furthermore benefits were largest for continuous cognitively engaging PA over several weeks [48]. In other words, what could correspond to continuous participation in LTS with physical movement and cognitively engaging content. Another study also found that LTS could be a relevant strategy to increase the general health-related PA level [50]. Unfortunately, however, not in such a way that participation in LTS leads to children being generally more physically active [51]. This was the result of a study including approximately 1200, 0th–6th grade children attending a “sport school” where the mandatory physical education (PE) program was increased from 2 to 6 weekly lessons over a 3-year period [51]. The children attending normal schools were offered the standard 2 PE lessons. They found no significant differences in PA levels during total time, PE, or recess between children attending sports schools and normal schools, respectively [51]. Sport schools children were more active than normal schools children during school time, but less active during leisure time. Which support that it is the organized sport which make the difference for the overall PA.

### 1.3. Socio-Economic Status and Executive Functions

Socio-economic status (SES) is a complex multi-dimensional construct [52], which is very often used in research literature in simplified designs [52] to measure the impact of e.g., parents educational level, income, neighborhood and school district has on the outcome measured. SES has previously shown to be of great importance for children’s academic performance in Denmark, e.g., showed by their score on “National Test”, a test all Danish children must perform ten times in the period of 2nd–8th grade [53]. In evaluations of the children’s academic performance [54], information about the children’s EF is implicit provided during the children’s abilities to control information processes and behavior [16,34]. And well-functioning EF are important for academic learning [16,52], which in general is significant for children growing up and attending school.

In Denmark there is a strong culture of participating in leisure time sport [55,56]. Nevertheless, no study, to our knowledge, have been conducted concerning the influence of LTS on EF. For this present study, data were collected weekly for 9 months making it possible to obtain more valid data and use a longitudinal design, which increases the strength of our analyzes, compared to many cross-sectional studies [57].

### 1.4. Objective

To investigate the influence of leisure time sport on executive functions in Danish children in 1st grade, and secondary to explore if socio-economy is a confounder for associations between leisure time sport and executive functions in Danish children in 1st grade.

This longitudinal study shows that LTS seems to have a positive effect on WM and seems to be a predictor for the BRIEF sub-score Initiate in 1st grade children. The adjustment with HPE data did not reveal any clinically relevant difference for LTS predicting EFs.

## 2. Materials and Methods

### 2.1. Study Design

This study is a sub-study nested within a cluster-randomized controlled trial (clinicaltrials.gov, NCT02488460) [49] in which the present report deals with outcome data in relation to LTS and EFs.

### 2.2. Participants and Setting

The participants came from 12 different elementary schools across 2 municipalities, Kolding and Svendborg, in the south of Denmark. All schools were co-educational schools, with gender integrated classes [49]. The data were collected from 1st grade children in the schoolyear 2012/2013 [58]

At baseline there were no significant differences between participants from Kolding and those from Svendborg regarding age and gender distribution, height, weight and BMI, as seen in Table 1; Baseline data.

#### 2.2.1. Inclusion Criteria

Children in 1st grade in a school from either of the 2 participating municipalities in the south part of Denmark [58]. Schools were eligible if they did not have a structured program that incorporated PA into the classroom [58].

#### 2.2.2. Exclusion Criteria

Physical disability and no written parental consent, as described in the study protocol of the main study [58].

#### 2.2.3. Invitation

The children and parents were invited by letter informing them about the study. The children, parents and teachers were invited to a meeting at the school where a full description of the study and the scientific background was presented.

## 3. Data Collection

### 3.1. Baseline and Follow up Measurements

During the first 2 months and the last 2 months of the school year, all participating children took part in baseline and follow-up assessments using BRIEF (Behavior Rating Inventory of Executive Function) and the Flanker/Reverse Flanker Task. Cognitive processes are challenging to measure since the influence of other cognitive processes, and other factors can affect cognitive functioning (e.g., low motivation, stress, etc.).

All data not automatically registered, was entered twice. When there were differences between datasets, the original values were checked, and the correct values entered.

### 3.2. Flanker/Reverse Flanker

The origin Flanker Task was developed in 1974 [59] and used letters; however, in the present study, we employed a modified version (the Flanker/Reverse Flanker Task) which used fish instead. Eriksen and Eriksen [59] explained that the purpose of the Flanker Task is to test if the subject is capable of the inhibitory control required to prevent the responses from running off “willy-nilly”. This task has been validated in previous studies [60] as sufficiently sensitive for detecting changes in cognitive function.

During the test, the child was instructed to “feed” the fish by pressing the arrow in the direction the target fish was facing (Figure 1). The test consisted of three conditions: standard flanker, reverse flanker, and mixed trials [61].

For our analysis, only mixed trials were used due to their complexity performing in these trials requires all 3 core EFs. These mixed trials were expected to be able to detect small differences between individuals and groups.

The Flanker/Reverse Flanker Task consisted of 45 trials; however, only 44 trials were included in the analysis due to possible delays at the start [58]. The child’s ability to focus and exclude interfering stimuli was tested for each of the 4 categories of the flanker task: (1) ‘congruent’ stimuli in which the flanking stimuli were identical to the target stimulus (12 tasks), (2) ‘incongruent’ stimuli, in which the flanking stimuli were the opposite of the target stimulus (16 tasks), (3) ‘no distractors’ with no flanking stimuli (8 tasks) and (4) ‘neutrals’, in which the flanking stimuli were neither identical nor opposite to the primary stimuli and were oriented vertically (8 tasks). The 44 trials were evenly divided between right and left as the correct answer. The stimulus presentation time was 1500 milliseconds (ms), the feedback interval was 1000 ms, and the interstimulus interval was 500 ms. For each trial, the response time was recorded, and four variables were calculated: (a) % accurate congruent answers, (b) % accurate incongruent answers, (c) the reaction time for correct congruent answers (ms), and d) the reaction time for correct incongruent answers (ms).

### 3.3. BRIEF Questionnaire

BRIEF-P (Behavior Rating Inventory of Executive Function—parent) is a questionnaire filled out by a parent or guardian, which assesses behaviors at home for children and adolescents aged 5–18 years for the following sub-scores: Initiate (the child’s ability to begin an activity and to independently generate ideas, responses, or problem-solving strategies) [62], WM, Plan/Organize, Organization and Materials (orderliness of work, play, and storage spaces) [62], Monitor (whether the child assesses their own performance to attain appropriate attainment of a goal) [62], Inhibit, Shift (cognitive flexibility), and Emotional Control (the impact of executive functions problems on emotional expression and assesses a child’s ability to modulate or control his or her emotional responses) [62]. Additionally, there are 3 overall scores: the metacognition index, the Behavior Regulation Index (the child’s ability to shift cognitive set and modulate emotions and behavior via appropriate inhibitory control) [62], and the Global Executive Composite X. We chose BRIEF because it screens for EF challenges in children and focuses on potential problems in the aforementioned areas of assessment.

If the parent did not return the questionnaire, they were reminded after 3–4 weeks through the school’s intranet [61].

### 3.4. Leisure-Time Sport

SMS-tracking (via the system SMS-track (https://sms-track.com (accessed on 6 December 2018)) [63] was used to assess LTS [61]. This was accomplished through weekly texting, asking the parents 3 questions (Appendix A) about how many times per week their child had participated in LTS the past week. Responses took form of a number between 0 and 8, where the number 8 was used when the child had participated in LTS more than 7 times [58]. We created 3 categories for the mean number of times a child participated in LTS per week:

Cat. 1, between 0.5 and 1.5; Cat. 2, between 1.5 and 2.5; Cat. 3, More than 2.5 times per week.

We performed analyzes to assess if children not doing any leisure time sport had different outcome than children doing any leisure time sport.

### 3.5. Socio-Economic Status

In the original RTC no overall significant differences between the two municipalities in age profile or inclusion in the workforce were found [49]. For the present study, the personal SES of parents from the Svendborg municipality was assessed using information from “Statistics Denmark” (https://dst.dk (accessed on 5 December 2018)), where SES-data on highest parental education level (HPE) for the parents living under the same roof as the child were obtained. Unfortunately, it was not possible to obtain personal SES of the parents in Kolding, due to the time went from the original study, where it was not collected.

## 4. Variables of Interest

The outcomes were results for the Flanker/Reverse Flanker Task and BRIEF. The exposures of interest were LTS and SES.

Regarding HPE, 5 categories were constructed (definitions translated according to the Ministry of Higher Education and Science [64]): Cat.1, researchers or higher education; Cat.2, bachelor from a university; Cat.3, bachelor in a profession such as physiotherapy; Cat.4, short education, e.g., crafts training or assistant in a shop; Cat.5, basic education or semi-skilled.

## 5. Covariates, Possible Moderators, and Confounders

The original randomized study revealed a difference in cognitive performance between the 4 groups [49], which consisted of (1) “Active math”, (2) “Control group” with no extra physical education (PE), (3) “Control group” with extra PE (-“6 PE lessons per week”) and (4) “Active math” with extra PE [58]. These 4 groups were incorporated into the analyses as nominal variables to adjust for the possible confounding intervention effect.

EF and math performance were selected based on evidence specifically linking inhibitory control with early mathematic ability [65]. The largest effects of exercise training have previously been found for executive functions in children [44] and elderly people [66], with math performance being significantly correlated with various measures of executive functioning [67].

The active math intervention consisted of math teaching that implemented PA in the classroom as a facilitating instrument. During the schoolyear the students received on average 6 math lessons of 45 min per week with physically active teaching. PA in this math intervention was defined as any bodily movement produced by skeletal muscles that resulted in increased energy expenditure. Teachers in the intervention schools attended a 4-day mandatory course, developed by the research team, on how to integrate active math into the Danish curriculum for mathematics in public schools.

In Svendborg municipality, the participating schools were all part of an existing intervention study that had been initiated in 2008 (the CHAMPS-study DK). This intervention consisted of four extra lessons of physical education (PE) each week, in addition to the two compulsory lessons, resulting in a total of 4.5 h extra per week. The primary focus in PE were the development of fundamental bodily skills and secondly sport-specific skills. The teachers aimed at making the environment fun and challenging, and with child-oriented playing, exercises, and small games.

To ensure that the extra PE lessons in Svendborg municipality did not bias the results of the study by Have et al. [58], randomization to the intervention was stratified by municipality.

Sex, age, height, and weight were measured at baseline and follow up and included as possible moderators and confounders.

## 6. Blinding

To avoid biases, all research assistants who administered the Flanker/Reverse Flanker Task (master and PhD students) were trained in the conduction of the test battery, to ensure uniform and valid data collection. All tests were performed at the participants’ schools—in classrooms, in the gym or in a small room with a maximum of 2 students present at the same time.

## 7. Analyzes

SES data for the families in Kolding were not available, so we performed 3 analyses: 1 for Kolding, 1 for Svendborg, and 1 for all participants together but without adjustment for SES.

### 7.1. Descriptives

Normality assumptions were checked graphically using histograms and qq-plots. For normally distributed data means and confidence intervals were calculated; for non-normally distributed data, 25%, 50% and 75% centiles were calculated.

Bivariate statistics were used for determining unadjusted differences over time. For normal distributed variables, the paired *t*-test was used; for non-normally distributed data, the Wilcoxon signed-ranks test was used.

### 7.2. Missing Data

In our exploration of the data, we compared baseline values for those lost to follow up and those with full follow up. Inclusion in the “full follow up” group required a full data set for BRIEF or Flanker/Reverse Flanker Task, or accelerometry, respectively (Table 2).

### 7.3. Multivariate Analyzes

Multivariate associations were assessed using mixed effects linear regression. To adjust for the clustering of schools and classes, these parameters were included as random variables in the analyses. The analyses were adjusted for relevant potential confounders (i.e., sex, age, baseline value of the variable, and HPE in Svendborg). The level of significance was set as *p* < 0.05.

To establish if there was a trend across categorical variables, we performed a test for trend whilst treating the variables as if they were continuous. The results of such a test indicate whether the categorical variables have a logical order, allowing a significant result to be interpreted with greater confidence.

Analyses were conducted using statistical software STATA version 15.0.

## 8. Results

Of all the parents, 90.7% gave consent for their child to participate in the study, corresponding to 505 out of 557 students from 26 different classes.

Missing data and lost to follow up (Table 2). LTS and EFs are presented in Table 3 (BRIEF) and Table 4 (Flanker/Reverse Flanker Task). Finally, we present results on whether SES is a confounder.

The analysis including all participants was performed due to HPE not being obtained for one of the municipalities, as mentioned earlier.

### 8.1. Missing Data and Lost to Follow Up

The response rates were as follows: BRIEF, 76.04% at baseline, 60% at follow up. Flanker/Reverse Flanker Task, 94.65% at baseline, 91.29% at follow up. The LTS overall response rate was 78.42%.

In Table 2, the differences between baseline data for those with missing data or lost to follow up are compared to those with full follow up using standard error (SE). Comparisons have been made via the unpaired *t*-test for normally distributed data and via the Wilcoxon Rank Sum for non-normally distributed data.

For the Svendborg participants, significant differences were found in 8 out of 11 BRIEF sub scores as well as the 3 overall scores: Behavioral Regulation Index, Metacognition Index, and Global Executive Composite (Table 2).

The participants with missing follow up data (non-full follow up) demonstrated the most improvement. This was found to be true for both Kolding and Svendborg participants. No difference in change over time was observed for the Flanker/Reverse Flanker Task between the 2 municipalities.

### 8.2. Leisure Time Sport and Executive Functions

#### 8.2.1. Overall

When looking at LTS or no LTS, we found a highly significant differences in between children doing no LTS and children doing LTS when looking at WM and Initiate (Table 3.). Further there was a clear dose response with LTS appearing to significantly improve WM in children in 1st grade throughout our study population (Table 3), with the effect reaching more than 8% for those participating in LTS between 0,5 and 1,5 times per a week to almost 20 percent for those participating between 2,5 times and up to 5.5 times per week.

Furthermore, we found that LTS was a significant predictor for Initiate cross our whole study population (Table 3) with the effect reaching nearly 8% for those participating in LTS between 0.5 and 1.5 times per week and almost 13% for those participating in LTS between 2.5 and up 5.5 times per week.

These results were found to be true regardless of how many times per week the child participated in LTS. No other logically significant effects were found for LTS. There did seem to be an effect on Organizing and Material, and on Monitor (Table 3), but such effects would not be logical, as the effects across groups were A-shaped and only a few of the groups were significantly associated with LTS.

The same result was found for the association between LTS and the Flanker/Reverse Flanker Task, as the effect here across groups only showed one significant result, but the curve across groups would be A- or U-shaped.

#### 8.2.2. Kolding

We found LTS to be a significant predictor for Initiate for Kolding participants, as in our overall study population (Table 3). Participating in LTS between 0.5 and 1.5 times per week in Kolding was also found to be a predictor for Emotional Control and Organization of Materials (Table 3). Participating in LTS more than 2.5 times per week in Kolding was found to be a significant predictor for Emotional Control and Behavior Rating Index (Table 3).

#### 8.2.3. Svendborg

For Svendborg participants, participating in LTS between 0.5 and 1.5 times per week was a predictor for Monitor (Table 3), whilst participating in LTS between 1.5 and 2.5 times per week was a predictor for Accuracy in the congruent trials on the Flanker/Reverse Flanker Task (Table 4).

### 8.3. Socio-Economy—A Confounder?

The adjustment with HPE data did not reveal any clinically relevant difference for LTS predicting EFs.

## 9. Discussion

### 9.1. Summary of Findings

To the authors knowledge, this is the first study examining the influence of LTS on EFs.

Our primary result indicated that if a child around the age of 7 were physically active in LTS every week (from 0.5 to more than 5 times per week), it would have a positive impact—up to an improvement of 20 percent for WM, which can be considered clinically relevant for praxis. EFs are linked to the prefrontal cortex in several studies [15,45], and being physically active has previously been shown to support neurogenesis in the prefrontal cortex [37,44]. However, regardless of the type of PA, the exact physiological reason for the effect of PA on EFs remains unclear as mentioned in the introduction.

Our results indicated different effects on different parts of the EFs measured using BRIEF [62]. Besides WM, our results also indicate a moderate effect of LTS on the sub-score Initiate, regardless of how many times per week they participated in LTS. Participating in LTS 0.5–1.5 times per week was found to be a predictor of Organization and Materials, whilst participating in LTS 0.5–2.5 times per week was a predictor for Monitor (Table 3). We did not observe any effect with those kids who practiced more sport, the reason for which it is considered to just be a coincidence.

We found LTS to be a significant predictor for Initiate for Kolding participants, as in our overall study population (Table 3). Participating in LTS between 0.5 and 1.5 times per week in Kolding was found to be a predictor for Emotional Control and Organization of Materials (Table 3). Participating in LTS more than 2.5 times per week in Kolding was also found to be a significant predictor for Emotional Control and Behavior Rating Index (Table 3). Again, it is not logical that, e.g., for Emotional Control, we found no significant effect in the group of children participating in sport 1.5–2.5 times per week. Such an effect was only found for those participating either less or more.

In Svendborg participating in LTS between 0.5 and 1.5 times per week was a predictor for Monitor (Table 3), whilst participating in LTS between 1.5 and 2.5 times per week was a predictor for Accuracy in the congruent trials in the Flanker/Reverse Flanker Task (Table 4).

It seems rather random how many times a week are needed to influence the different parts of the EFs in the examples above. More specific studies designed for measuring this phenomenon are recommended. Nevertheless, the results could be random, or they could be impacted by more cognitive strategies involved in participating in LTS, such as following rules, clearing up gear and being a good friend and team player.

We found a significant effect on WM, and we assumed in the introduction that cognitive flexibility was dependent on WM and the ability to inhibit, but why did we then not see an effect on cognitive flexibility? In the BRIEF manual, the sub-scores -“Shift”- and -“inhibition” are part of “behavioral regulation” [62], whilst WM is part of “metacognition” [62]. One could argue that behavioral regulation (defined as: Emotional Control, Shift and Inhibition in BRIEF) is a more feminine values than metacognition (defined as Monitor, Organization and Materials, Plan/Organize, WM and Initiate in BRIEF) and thus reflects values which are more measurable and more easily recognized and remembered by the parent who fills out the BRIEF questionnaire. So, are BRIEF test and modified Flanker/Reverse Flanker Task responsive enough to capture the more feminine values?

We still believe the skills of WM and inhibition are required to possess cognitive flexibility, because one must be able to hold information in their mind (WM) and to inhibit an “old” way of thinking to initiate a new way of thinking about a specific dilemma, problem, or situation.

Our results for WM and LTS are in accordance with other studies. A recent RCT and metanalysis [59] found significant results for chronic exercise interventions in curricular or sports and PA settings. Even though the effect sizes were small, it is a promising analysis; furthermore, the results are supported by other studies such as that conducted by Alesi et al. [14] where improvements in WM were shown from a 6-month football training program in children aged 8.8 years old [14].

No studies regarding sports other than football can be found. In our study, no adjustments have been made regarding sport type, but football is the sport type in Denmark with the most participants [68]. No study indicating that LTS does not improve WM can be found, and our results for LTS and EFs represent a positive “stepping-stone” in research regarding the influence of LTS on WM.

In contrast to our findings Sjöwall et al. [69] explored the causal relationship between PA and WM in a 2-year trial and did not find an association. However, other studies examining the exercise–cognition relationship have revealed positive associations between PA and cognitive performance [48,70,71].

In a metanalysis conducted by De Greef et al. [48] on RCTs and cohort studies [48], with 6 out of 12 longitudinal cohort studies investigating subdomains of EFs, a small to moderate positive effect was found between longitudinal PA programs and WM (and EFs in general) [48]. They concluded that PA over several weeks will increase the chance of improving EFs and academic performance even more so if the intervention includes cognitively challenging PA [48]. From a longitudinal perspective, duration in weeks had no significant influence on the effect of PA on EFs [48].

An RCT study conducted by Hsieh et al. [70], showed that a higher amount of PA resulted in superior WM in children aged 8–11 years, indicated by both behavioral and neuroelectric measures [70]. However, whether it is the physiological processes of PA or the social part of PA triggering a causal relationship is not known.

In Svendborg, the significant results for the effect of LTS on EFs were inconsistent across the BRIEF sub-scores, except for WM (as in Kolding). This means that HPE is not affecting our results from the mixed effect regression analysis. One school year (9 months) could be thought of as a rather short period of time to show an effect from HPE on PA or LTS, which could then affect EFs.

Furthermore, it is a limitation that it is not known whether it is PA per se that is the factor influencing EF, or e.g., increased parental support of any activity.

### 9.2. Methodological Considerations

The study design chosen for this present study treats data as prospective longitudinal outcome data in relation to LTS.

A limitation could be that we statistically we performed multiple tests and did not adjust for this. Thus the results should be looked upon hypothesis generating more than results showing a causal relationships. The results reveal that we have full follow up for those who did well in the test, whilst those who did not perform that well were missing at follow up. This could have influenced our results, increasing the risk of a type one error—a false-positive result. Furthermore, for the group for whom we have full follow up, the baseline BRIEF values were lower meaning they did better on the test, and the confidence intervals were narrower, meaning less variation in the sample. In addition, the differences in the number of participants between the two groups (full follow up *n* = 128–132; lost to follow up or missing *n* = 42–45) also contributes to our understanding that the results could be overestimated.

The Flanker/Reverse Flanker Task enables an objective measure of EFs, specifically the child’s ability to focus on particular stimuli while ignoring others; as such, it has been used in a number of previous studies to assess EFs [31,60,72].

During the test, the child becomes familiarized with the test through 21 practice trials, followed by 44 “real” trials divided equally between the four flanker conditions. One concern could be that a total of only 10 tasks for each condition is too low for drawing solid conclusions, whilst also considering the significant variability observed in children in general. In comparison 200 tasks have been used in the Flanker/Reverse Flanker Task in other studies [73,74]. Thus, this could be a limitation in our study.

Although BRIEF is often used as an assessment tool in research [58,75] it has only been examined to a very limited extend for its psychometric abilities in relation to children with no disabilities. McAuley et al. [75] argue that BRIEF is better suited to finding behavioral problems than measuring EFs; as such, they advise and recommend that it is important for future studies to verify the validity of BRIEF by comparing the questionnaire to tasks that require EF in more complex contexts. This was not done in our study and possibly constituting another limitation.

Collecting data on LTS from SMS-track was assumed to be easy and convenient for the parents. However, in cleaning the raw data, it was observed that several parents wrote statements for responses (instead of numbers as they were supposed to report) regarding their inconvenience answering consecutive SMS’s over such a long period. All parents were informed about the duration of the main study [58], but still, it is imagined that they were hampered by the repeated requests for their attention to the study. Nevertheless, it is a strength of the study regarding recall biases. Quite a lot of statistical tests were performed, so there is a risk that the few statistically significant findings are chance findings. We used a significance level of *p* = 0.05 and believe it to be appropriate for the results and interpretation thereof. Correction for multiple analyses were not performed, it could have been done as hypothesis generating.

## 10. Conclusions

This longitudinal study shows that LTS seems to have a positive effect on WM and seems to be a predictor for the BRIEF sub-score Initiate in 1st grade children. The adjustment with HPE data did not reveal any clinically relevant difference for LTS predicting EFs.

In perspective, the results of this study will add to current evidence for the relationship between LTS and EFs in 1st grade children in Denmark.

Further research should aim to also take the social aspect of LTS into account. In addition, one should be critical regarding the psychometric properties of tests measuring EFs, when designing new studies on this subject.

## Figures and Tables

**Figure 1 children-09-01458-f001:**
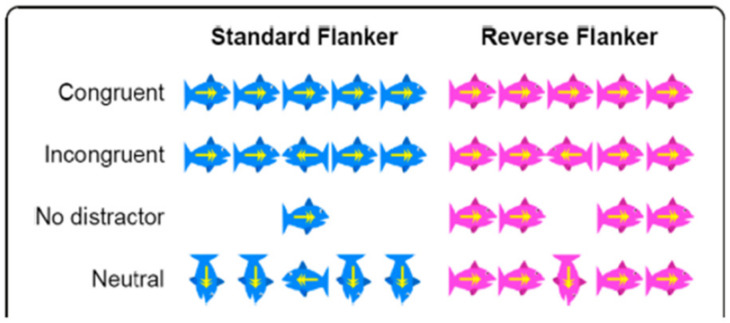
Examples of the stimuli in the modified Flanker test of executive functions showing blue fish’s: Standard flanker test (target stimulus = middle fish) and pink fish’s: Reverse flanker test (target stimulus = ‘non-middle’ fish). Borrowed from Have et al. [58].

**Table 1 children-09-01458-t001:** Demographic Data. Baseline data. Study Population. Data is presented as mean (CI) for normal distributed data and by medians and 25% and 75% centiles for non-normally distributed data.

	Kolding	Svendborg	All
	Mean (95% CI) n	Mean (95% CI) n	Mean (95% CI) n
	**Baseline**	**Baseline**	**Baseline**
**Variable/Gruppe**			
n	187–232	177–230	384–505
Age (year)	7.2 (7.2–7.3) n = 230	7.2 (7.2–7.2) (n = 229)	7.2 (7.2–7.3) (n = 495)
Gender (%boys)	47.41% n = 232	51.74% (n = 230)	49.50% (n = 505)
Highest Parental Education *Denoted as social categori*		Soc.cat 1: 3.46% Soc.cat 2: 0% Soc.cat 3: 27.71% Soc.cat 4: 37.23% Soc.cat 5: 27.71% *Missing: 3.90%*	
ANTHROPOMETRIC MEASURES			
Height (cm)	126.9 (126.2–127.6) n = 224	126.8 (126.1–127.5) (n = 227)	126.7 (126.2–127.2) (n = 485)
Weight (kg)	25.9 (25.3–26.6) n = 224	25.8 (25.2–26.3) (n = 227)	25.7 (25.3–26.1) (n = 485)
BMI (kg/m^2^)	16.1 (15.7–16.3) n = 224	15.9 (15.7–16.2) (n = 227)	15.9 (15.7–16.1) (n = 485)
FLANKER			
Congruent reaction time (ms)	1381.51 (1337.37–1425.65) (n = 225)	1374.38 (1326.51–1422.25) (n = 220)	1374.94 (1343.01–1406.86) (n = 478)
Incongruent reaction time (ms)	1759.09 (1703.25–1814.93) (n = 225)	1755.17 (1698.45–1811.90) (n = 220)	1751.68 (1712.90–1790.46) (n = 478)
Congruent accuracy (%)	95.00 (94.00–96.00) (n = 225)	93.56 (91.78–95.34) (n = 220)	93.93 (92.88–94.99) (n = 478)
Incongruent accuracy (%)	77.61 (75.74–79.48) (n = 225)	75.00 (72.67–77.33) (n = 220)	76.27 (74.81–77.73) (n = 478)
BRIEF			
Inhibition	15.12 (14.63–15.61) (n = 187)	14.85 (14.36–15.35) (n = 177)	15.04 (14.70–15.38) (n = 384)
Shift	11.71 (11.29–12.15) (n = 187)	11.73 (11.32–12.14) (n = 177)	11.80 (11.51–12.09) (n = 384)
Emotional control	16.44 (15.86–17.03) (n = 187)	15.77 (15.20–16.34) (n = 177)	16.17 (15.77–16.57) (n = 384)
Initiate	12.51 (12.15–12.87) (n = 187)	12.42 (12.03–12.82) (n = 177)	12.5 (12.24–12.76) (n = 384)
Working memory	16.32 (15.69–16.95) (n = 187)	15.94 (15.34–16.53) (n = 177)	16.21 (15.78–16.63) (n = 384)
Plan and organization	17.67 (16.94–18.41) (n = 187)	17.04 (16.32–17.76) (n = 177)	17.48 (16.98–17.97) (n = 384)
Organization of materials	12.29 (11.85–12.72) (n = 187)	12.01 (11.56–12.45) (n = 177)	12.19 (11.88–12.49) (n = 384)
Monitor	12.82 (12.34–13.31) (n = 187)	12.63 (12.19–13.08) (n = 177)	12.79 (12.47–13.11) (n = 384)
Behavioral regulation index (BRI) (overall)	43.20 (41.90–44.50) (n = 187)	42.21 (40.94–43.49) (n = 177)	42.92 (42.03–43.81) (n = 384)
Metacognition index (MCI) (Overall)	70.16 (67.53–72.78) (n = 187)	68.21 (65.53–70.89) (n = 177)	69.61 (67.79–71.43) (n = 384)
Global executive composite (GEC)	112.20 (108.16–116.24) (n = 187)	108.75 (104.43–113.07) (n = 177)	111.20 (108.34–114.05) (n = 384)

**Table 2 children-09-01458-t002:** Comparison of baseline characteristics for those with follow up and those with non-full follow up. Normally distributed data is presented with mean (sd) (95% CI) and n. Non-normally distributed data is presented with medians (95% CI) and n, or with percentages. Between group differences are presented as crude values with (95% CI) or percentages an *p*-values. * The star denotes between group differences in non-normally distributed data.

	Full Follow-Up vs. Non Full Fullow-Up							
Variable/Group	Kolding				Svendborg			
	Baseline Characteristics for Those with Full Follow up	Baseline Characteristics for Those with Missing Data or Lost to Follow up	Differences (SE) (95% CI)	*p* for Difference	Baseline Characteristics for Those with Full Follow up	Baseline Characteristics for Those with Missing Data or Lost to Follow up	Differences (SE) (95% CI)	*p* for Difference
ANTHROPOMETRIC MEASURES								
Age (year)	7.25 (7.21–7.30) n = 215	7.15 (6.97–7.33) n = 9	−0.11 (−0.35–0.13)	0.383	7.20 (7.16–7.24) n = 216	7.14 (6.92–7.36) n = 11	−0.06 (−0.25–0.13)	0.518
Gender (%boys)	46.12%	45.69%	0.43% percentage point	0.954	50.65%	47.19%	3.46 percentage point	0.850
Height (cm)	126.87 (126.12–127.62) n = 215	127.31 (126.12–132.53) n = 9	0.44 (−3.33–4.21)	0.818	126.83 (126.05–127.52) n = 216	125.99 (124.14–127.85) n = 11	−0.84 (2.80–1.12)	0.377
Weight (kg)	25.87 (25.23–26.51) n = 215	27.29 (22.98–31.60) n = 9	1.42 (−1.79–4.63)	0.384	25.78 (25.17–26.38) n = 216	25.49 (23.93–27.05) n = 11	−0.29 (−1.93–1.36)	0.715
BMI (kg/m^2^)	15.98 (15.71–16.25) n = 215	16.70 (15.00–18.40) n = 9	0.72 (−0.64–2.07)	0.298	15.94 (15.68–16.21) n = 216	16.05 (15.21–16.88) n = 11	0.10 (−1.10–1.31)	0.865
FLANKER								
Congruent reaction time (ms)	1379.01 (1332.66–1425.36) n = 209	1414.17 (1261.13–1567.21) n = 16	35.16 (−136.90–207.22)	0.688	1412 (CI 1342.68–1468.83) n = 129	1262.55 (CI 1076.42–1372.16) n = 42	*149.45	0.9213
Incongruent reaction time (ms)	1754.29 (1695.64–1812.94) n = 209	1821.78 (1630.82–2012.74) n = 16	67.49 (−150.09–285.07)	0.542	1754.13 (CI 1704.89–1821.52) n = 129	1711.90 (CI 1429.14–1845.42) n = 42	*42.23	0.4815
Congruent accuracy (%)	100 (CI 91.67–100) n = 111	100 (CI 91.67–100) n = 72	*0	0.564	100 (CI 100–100) n = 129	100 (CI 91.67–100) n = 42)	*0	0.1162
Incongruent accuracy (%)	81.25 (CI 75–81.25) n = 111	81.25 (CI 75–87.5) n = 72	*0	0.841	81.25 (CI75–81.25) n = 129	71.88 (CI 62.5–81.25) n = 42	*9.37	0.162
BRIEF								
Inhibition	14 (CI 14–15) n = 114	15 (CI 14–16.82) n = 72	*1	0.156	14 (CI 13.78–15) n = 131	17 (CI 15–18) n = 45	*3	0.000
Shift	11 (CI 10–12) n = 114	12 (CI (11–12.82) n = 72	*1	0.441	11.45 (11.00–11.91) n = 132	12.53 (11.65–13.42) n = 45	1.08 (0.15–2.00)	0.023
Emotional control	16 (CI 15–17) n = 114	16 (CI 16–18.40) n = 73	*0	0.361	16 (CI 14–16) n = 132	17 (CI 15–18.61) n = 45	*1	0.077
Initiate	12.28 (11.82–12.74) n = 114	12.86 (12.29–13.43) n = 73	0.58 (−0.15–1.31)	0.118	12 (CI 12–13) n = 132	13 (CI 11–13) n = 45	*1	0.507
Working memory	16 (CI 14–17) n = 114	17 (CI 16–18) n = 72	*1	0.180	15 (CI 14–16) n = 132	17 (CI 16–19) n = 45	*2	0.000
Plan and organization	17 (CI 16–18) n = 111	19 (CI 17–20) n = 70	*2	0.100	17 (CI 16–18) n = 129	19 (CI 17–21) n = 42)	*2	0.006
Organization of materials	12.24 (11.70–12.77) n = 114	12.37 (11.62–13.12) n = 73	0.13 (−0.76–1.03)	0.770	11.85 (11.32–12.38) n = 132	12.47 (11.62–13.31) n = 45	0.62 (−0.40–1.64)	0.2347
Monitor	13 (CI 12–14) n = 114	13 (CI 12–14) n = 71	*1	0.418	12 (CI 12–13) n = 132	14 (CI 13–15) n = 45	*2	0.001
Behavioral regulation index (BRI) (overall)	42.78 (41.25–44.31) n = 114	44.47 (42.43–46.51) n = 72	1.69 (−0.81–4.19)	0.184	41 (CI 39–43) n = 131	45 (CI 42–49.61) n = 45	*4	0.003
Metacognition index (MCI) (Overall)	71.29 (68.90–73.68) n = 111	74.37 (71.09–77.65) n = 70	3.08 (−0.87–7.03)	0.125	68.75 (66.65–70.85) n = 129	76.29 (71.89–80.68) n = 42	7.53 (3.12–11.95)	0.001
Global executive composite (GEC)	114.15 (110.60–117.71) n = 111	118.73 (113.96–123.50) n = 70	4.58 (−1.25–10.40)	0.123	110.34 (107.24–113.43) n = 128	122.05 (115.36–128.73) n = 42	11.71 (3.32) (5.15–18.27)	0.001

**Table 3 children-09-01458-t003:** BRIEF—Results from Kolding & Svendborg and all together. Results are presented with mean, *p*-values and 95% confidence interval (CI). Grey boxes: Significant values. Categories for LTS: Cat.0: =0 (not shown, reference cat. = 0), Cat.1: ≥0.5–<1.5, Cat.2: ≥1.5–<2.5, Cat.3: ≥2.5–<5 times per week.

	Kolding	Svendborg	All
	LTS Mean (*p*-Value) (95% CI)	LTS Mean (*p*-Value) (95% CI)	LTS Mean (*p*-Value) (95% CI)
BRIEF	(n = 101)	(n = 107)	(n = 213)
Inhibition	(1) −0.56 (*p* = 0.399) (−1.86–0.74) (2) −0.63 (*p* = 0.385) (−2.06–0.80) (3) −1.90 (*p* = 0.126) (−4.33–0.53)	(1) −0.74 (*p* = 0.236) (−1.95–0.48) (2) −0.11 (*p* = 0.883) (−1.54–1.33) (3) −0.33 (*p* = 0.770) (−2.62–1.95)	(1) −0.59 (*p* = 0.190) (−1.46–0.29) (2) −0.36 (*p* = 0.476) (−1.34–0.63) (3) −0.91 (*p* = 0.283) (−2.58–0.75)
Shifting	(1) −0.96 (*p* = 0.146) (−2.15–0.32) (2) −1.36 (*p* = 0.051) (−2.72–0.01) (3) −1.73 (*p* = 0.155) (−4.12–0.65)	(1) −0.01 (*p* = 0.985(−1.03–1.01) (2) 0.41 (*p* = 0.510) (−0.81–1.64) (3) 0.28 (*p* = 0.770) (−1.61–2.17)	(1) −0.23 (*p* = 0.565) (−1.02–0.56) (2) −0.48 (*p* = 0.292) (−1.36–0.41) (3) −0.51 (*p* = 0.510) (−2.02–1.00)
Emotional control	(1) −1.59 (*p* = 0.048) (−3.17–−0.01) (2) −1.22 (*p* = 0.173) (−2.97–0.53) (3) −3.45 (*p* = 0.028) (−6.53–−0.37) Test for trend −0.63 (*p* = 0.099) (−1.39–0.12)	(1) −0.77 (*p* = 0.291) (−2.19–0.65) (2) 0.36 (*p* = 0.675) (−2.02–1.31) (3) 0.07 (*p* = 0.958) (−2.60–2.74)	(1) −0.97 (*p* = 0.071) (−2.03–0.08) (2) −0.62 (*p* = 0.304) (−1.80–0.56) (3) −1.42 (*p* = 0.174) (−3.46–0.62)
Initiate	(1) −1.61 (*p* = 0.010) (−2.83–0.39) (2) −2.49 (*p* = 0.000) (−3.86–−1.13) (3) −3.11 (*p* = 0.009) (−5.45–−0.77) Test for trend −1.09 (*p* = 0.000) (−1.66–−0.52)	(1) −0.07 (*p* = 0.189) (−1.80–0.36) (2) −0.91 (*p* = 0.161) (−2.17–0.36) (3) 0.46 (*p* = 0.663) (−2.51–1.60)	(1) −0.96 (*p* = 0.020) (−1.78–−0.15) (2) −1.45 (*p* = 0.002) (−2.36–−0.54) (3) −1.62 (*p* = 0.043) (−3.18–−0.05) Test for trend −0.61 (*p* = 0.002) (−0.99–−0.22)
Working memory	(1) −1.95 (*p* = 0.008) (−3.39–−0.50) (2) −2.01 (*p* = 0.011) (−3.66–−0.47) (3) −4.46 (*p* = 0.001) (−7.20–−1.73) Test for trend −1.03 (*p* = 0.003) (−1.71–−0.36)	(1) −1.31 (*p* = 0.040) (−2.56–−0.06) (2) −1.51 (*p* = 0.045) (−3.00–−0.03) (3) −2.56 (*p* = 0.031) (−4.88–−0.24) Test for trend −0.74 (*p* = 0.019) (−1.35–−0.12)	(1) −1.36 (*p* = 0.005) (−2.31–−0.41) (2) −1.85 (*p* = 0.001) (−2.92–−0.78) (3) −3.16 (*p* = 0.001) (−4.97–−1.35) Test for trend −0.90 (*p* = 0.000) (−1.35–−0.45)
Plan & Organization	(1) 0.78 (*p* = 0.490) (−1.44–3.00) (2) 1.29 (*p* = 0.303) (−1.17–3.76) (3) −1.34 (*p* = 0.539) (−5.62–2.94)	(1) −1.58 (*p* = 0.114(−3.55–0.38) (2) −0.22 (*p* = 0.849) (−2.54–2.09) (3) −0.50 (*p* = 0.788) (−4.18–3.17)	(1) −0.37 (*p* = 0.628) (−1.85–−1.12) (2) −0.06 (*p* = 0.939) (−1.73–1.60) (3) −0.82 (*p* = 0.575) (−3.68–2.04)
Organising og materials	(1) −1.33 (*p* = 0.028) (−2.52–−0.14) (2) −0.95 (*p* = 0.165) (−2.29–0.39) (3) −1.83 (*p* = 0.120) (−4.14–0.48) Test for trend −0.36 (*p* = 0.222) (−0.94–0.22)	(1) −0.62 (*p* = 0.272(−1.72–0.49) (2) −0.32 (*p* = 0.629) (−1.62–0.98) (3) −0.63 (*p* = 0.559) (−2.73–1.48)	(1) −0.90 (*p* = 0.030) (−1.72–−0.09) (2) −0.74 (*p* = 0.115) (−1.65–0.18) (3) −1.02 (*p* = 0.205) (−2.60–0.56)
Monitor	(1) −0.72 (*p* = 0.259) (−1.98–0.53) (2) −0.24 (*p* = 0.735) (−1.66–1.17) (3) −2.02 (*p* = 0.096) (−4.40–0.36)	(1) −1.61 (*p* = 0.004) (−2.72–−0.50) (2)−1.42 (*p* = 0.032) (−2.71–−0.12) (3) −0.68 (*p* = 0.519(−2.77–1.40) Test for trend −0.37 (*p* = 0.191) (−0.93–0.18)	(1) −1.00 (*p* = 0.019) (−1.84–−0.17) (2) −0.98 (*p* = 0.043) (−1.92–−0.03) (3) −0.99 (*p* = 0.222) (−2.58–0.60) Test for trend −0.34 (*p* = 0.096) (−0.74–0.06)
Behavior regulation index	(1) −2.97 (*p* = 0.071) (−6.19–0.25) (2) −3.11 (*p* = 0.086) (−6.68–0.45) (3) −6.28 (*p* = 0.048) (−12.50–−0.05) Test for trend −1.49 (*p* = 0.055) (−3.00–0.03)	(1) −1.33 (*p* = 0.390) (−4.37–1.71) (2) 0.22 (*p* = 0.903) (−3.37–3.82) (3) 0.18 (*p* = 0.951) (−5.58–5.94)	(1) −1.74 (*p* = 0.117) (−3.92–0.44) (2) −1.11 (*p* = 0.376) (−3.56–1.34) (3) −2.61 (*p* = 0.224) (−6.83–1.60)
Metacogntiion index	(1) −1.48 (*p* = 0.695) (−8.86–5.90) (2) 0.79 (*p* = 0.850) (−7.45–9.03) (3) −9.56 (*p* = 0.187) (−23.78–4.66)	(1) −4.61 (*p* = 0.228) (−12.10–2.88) (2)−1.74 (*p* = 0.699) (−10.56–7.08) (3) −3.55 (*p* = 0.625) (−17.80–10.70)	(1) −2.78 (*p* = 0.308) (−8.11–2.56) (2) −2.23 (*p* = 0.466) (−8.23–3.76) (3) −6.18 (*p* = 0.239) (−16.48–4.18)
Generel Executive Functions	(1) −1.72 (*p* = 0.763) (−12.91–9.47) (2) 2.02 (*p* = 0.751) (−10.47–14.52) (3) −14.46 (*p* = 0.190) (−36.06–7.14)	(1) −5.50 (*p* = 0.342) (−16.85–5.85) (2) 0.04 (*p* = 0.995) (−13.36–13.45) (3) −3.21 (*p* = 0.771) (−24.81–18.38)	(1) −3.14 (*p* = 0.446) (−11.22–4.93) (2) −1.02 (*p* = 0.827) (−10.11–8.08) (3) −8.61 (*p* = 0.279) (−24.20–6.98)

**Table 4 children-09-01458-t004:** FLANKER—Results from Kolding & Svendborg and all together. Results are presented with mean, *p* values, 95% CI. Reaction time (ms) for correct congruent answers. Reaction time (ms) for correct incongruent answers. % Accurate congruent answers. % Accurate incongruent answers. Grey boxes: Significant values. Categories for LTS: Cat.1: ≥ 0.5–<1.5, Cat.2: ≥1.5–<2.5, Cat.3: ≥2.5–<5.5 times per week.

	Kolding	Svendborg	All
	LTS Mean (*p*-Value) (95% CI)	LTS Mean (*p*-Value) (95% CI)	LTS Mean (*p*-Value) (95% CI)
FLANKER	(n = 176)	(n = 179)	(n = 360)
Flanker reactiontime congruent trial block 3	(1) −70.99 (*p* = 0.222)(−185.04–43.05) (2) −58.46 (*p* = 0.405)(−196.03–79.12) (3) 158.69 (*p* = 0.197)(−82.53–399.91)	(1) 38.83 (*p* = 0.480)(−69.03–146.68) (2) 63.05 (*p* = 0.340)(−66.33–192.42) (3) 21.22 (*p* = 0.846)(−192.35–234.79)	(1) −3.94 (*p* = 0.917)(−78.47–70.58) (2) 21.43 (*p* = 0.641)(−68.67–111.53) (3) 91.95 (*p* = 0.256)(−66.61–250.50)
Flanker reactiontime incongruent trial block 3	(1) −2.83 (*p* = 0.962)(−120.13–114.47) (2) −105.54 (*p* = 0.138)(−244.96–33.89) (3) 6.87 (*p* = 0.957)(−241.45–255.18)	(1) −29.26 (*p* = 0.611)(−141.88–83.36) (2) −0.72 (*p* = 0.992)(−137.21–135.77) (3) 54.04 (*p* = 0.643)(−174.68–282.75)	(1) 6.43 (*p* = 0.873)(−72.57–85.43) (2) −25.28 (*p* = 0.600)(−119.74–69.19) (3) 45.45 (*p* = 0.597)(−122.80–213.70)
Flanker Accuracy congruent trial block 3	(1) −0.06 (*p* = 0.957)(−2.38–2.25) (2) 0.65 (*p* = 0.642)(−2.08–3.38) (3) −1.77 (*p* = 0.484)(−6.72–3.18)	(1) 1.69 (*p* = 0.128)(−0.48–3.87) (2) 3.48 (*p* = 0.010)(0.84–6.11) (3) 2.84 (*p* = 0.207)(−1.57–7.25) Test for trend 1.40 (*p* = 0.014)(0.28–2.51)	(1) 0.94 (*p* = 0.234)(−0.61–2.49) (2) 1.49 (*p* = 0.112)(−0.34–3.32) (3) 0.82 (*p* = 0.628)(−2.49–4.13)
Flanker Accuracy incongruent trial block 3	(1) 2.81 (*p* = 0.256)(−2.03–7.64) (2) 2.81 (*p* = 0.336)(−2.91–8.53) (3) 3.05 (*p* = 0.565)(−7.34–13.44)	(1) 4.29 (*p* = 0.055)(−0.09–8.67) (2) 2.60 (*p* = 0.335)(−2.69–7.89) (3) 6.54 (*p* = 0.150)(−2.37–15.46)	(1) 2.96 (*p* = 0.068)(−0.22–6.13) (2) 2.05 (*p* = 0.284)(−1.70–5.81) (3) 5.04 (*p* = 0.147)(−1.78–11.86)

## Data Availability

The data from this study contain potentially sensitive information. The data are available upon request; however, a data transfer agreement must be established to cover the transfer of datasets in accordance with the Danish Act on Processing of Personal Data. Interested researchers may direct data queries to servicedesk@sdu.dk.

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
