# Peer review of "The Effect of Leisure Time Sport on Executive Functions in Danish 1st Grade Children"

_children, 2022, doi:10.3390/children9101458_

Round 1

Reviewer 1 Report

Thank you for allowing me to review this manuscript on leisure time sport on executive functions of Danish children. I thought it was well written and easy to follow. While the findings were not all positive, it still contributes to the literature and has a large number of participants with a lot of reported data. Here are a few items I believe should be clarified/improved:

1. The data are from 2012-2013. That is a bit concerning, as they are almost a decade old. Do you see issues with this? Please justify in the text.

2. Section 3.4 states four categories but only lists three. Is there a fourth not listed?

3. Section 3.5.2 is confusing. I believe the authors should provide a better explanation of how/where this fits. What is active math? What did that consist of? Did the teachers have training? What did the PE lessons consist of? How long were they?

4. I am confused on the missing follow-up data. How are you able to use these participants/data?

5. Lines 316-317 seem to contradict the previous paragraphs. Please elaborate.

6. 4.2.3 Did any parents report NO LTS? Should you compare data from those participants to those with some/any LTS? Since it is stated that the specific amount of LTS did not matter, perhaps NO LTS matters either?

6. How accurate is the BRIEF data you collected? The questions raised in limitations do, in fact, seem very limiting.

Best of luck with the future of this manuscript!

Author Response

Response

Reviewer 1

Thank you very much for your great questions regarding our manuscript “The effect of leisure time sport on executive functions in Danish 1st grade children”.

We'll start by making it clear that references to certain sections can confuse in that there is a shift in the process at Children in "Round 1" in some of the numbers on the episodes and therefore they don't necessarily fit. We’ve fixed it but be aware.  

The data are from 2012-2013. That is a bit concerning, as they are almost a decade old. Do you see issues with this? Please justify in the text.

1) We are not concerned about the data being from 2012-2013. The issue is still relevant today.  Physical inactivity among first graders is unfortunately still an increasing problem in todays society in Denmark(1). Still influencing cognitive function relevant for executive function(2).

Section 3.4 states four categories but only lists three. Is there a fourth not listed?

2) Regarding your question 2 about section 3.4, it is a mistake. I’m sorry you got confused. There are no fourth category listed.

Section 3.5.2 is confusing. I believe the authors should provide a better explanation of how/where this fits. What is active math? What did that consist of? Did the teachers have training? What did the PE lessons consist of? How long were they?

3) There was no section 3.5.2.in the original draft you have been reading, but I believe it must be, what is now section “5. Covariates, possible moderators, and confounders”, where the randomization in the original study(3) is referred to.

The following is added the manuscript:

The active math intervention consisted of math teaching that implemented PA in the classroom as a facilitating instrument. During the schoolyear the students received on average 6 math lessons of 45 min per week with physically active teaching. PA in this math intervention was defined as any bodily movement produced by skeletal muscles that resulted in increased energy expenditure. Teachers in the intervention schools attended a 4-day mandatory course, developed by the research team, on how to integrate active math into the Danish curriculum for mathematics in public schools. 

In Svendborg municipality, the participating schools were all part of an existing intervention study that had been initiated in 2008 (the CHAMPS-study DK). This intervention consisted of four extra lessons of physical education (PE) each week, in addition to the two compulsory lessons, resulting in a total of 4.5 h extra per week. The primary focus in PE were the development of fundamental bodily skills and secondly sport-specific skills. The teachers aimed at making the environment fun and challenging, and with child-oriented playing, exercises, and small games.

To ensure that the extra PE lessons in Svendborg municipality did not bias the results of the study by Have et al.(3), randomization to the intervention was stratified by municipality.

I am confused on the missing follow-up data. How are you able to use these participants/data?

4) The purpose of using the missing follow-up data is to make the outcome more realistic. In the Non-full follow-up group were many of those who scored the least on BRIEF (which is assumed to be representative of the age group), and perhaps they have a different development curve, perhaps a flatter curve over the year - physically or cognitively. We can only guess about that. But the integration of their data is important for the results to reflect reality as much as possible. The group where we have full follow-up is the group that performed best on the cognitive tests, even before the intervention. This means that the results will be more positive than reality is, as it is the "good ones" that take up the most space in the analysis.

Lines 316-317 seem to contradict the previous paragraphs. Please elaborate.

5) (now line 335-336) This has been corrected – I’m sorry about the confusion! It was not “se” we used but 95% CI for the normally distributed data and Fisher exact test was used to assess there was a difference in sex distribution between the children with missing those without missing.

4.2.3 Did any parents report NO LTS? Should you compare data from those participants to those with some/any LTS? Since it is stated that the specific amount of LTS did not matter, perhaps NO LTS matters either?

6) (now section 8.2.3) This is a very good question – thank you for bringing it up, we have performed the analyzes, and have not found any difference between doing any LTS and no (zero) LTS, and we have included it as a sensitivity analysis.                                                                                               

How accurate is the BRIEF data you collected? The questions raised in limitations do, in fact, seem very limiting.

7) The BRIEF data collected is rated to be accurate, valid and reliable as much as the test allows. The parents were instructed on how to fill out the forms and had access to ask questions of the research team if necessary. The response rate was as follows for BRIEF; 76.04% at baseline and 60% at follow up. Which we considered reasonable for the analysis, though we have commented on it in the discussion, and included it as a limitation.

Thank you again for using time on our manuscript.

Reviewer 2 Report

This is a longitudinal study which investigated the influence of LTS on EFs  and explored if SES is a confounder for associations between LTS and EF in Danish 1st graders. You conducted appropriate design, data collection, and data analysis. The conclusions based on the results are practical and meaningful for the professionals in the area of physical activity and health. The manuscript is well organized. Except for the strengths, there are some minor concerns: 

1. Missed punctuations in Line 213 after "[43]" 

2. Misprint in Line 97: "2th" should be "2nd"

3. I guess the paragraph from line 112 to line 115 is the hypothesis of the study. Suggest to reorganize the language.

4. In the section "3.4 Leisure-time sport". The parent/guardian reported the number of times leisure-time sport participation, however "hours" are used when you analyzed the data. Please clarify it. 

Author Response

Response

Reviewer 2

Thank you very much for your great questions regarding our manuscript “The effect of leisure time sport on executive functions in Danish 1st grade children”.

I’ll start by a correction of the sections numbers. Unfortunately, there were a mistake in section 2 and 3 in the draft you have been reading. It is now corrected.

  1. Missed punctuations in Line 213 after "[43]"

         1) Have been corrected in the manuscript.

  1. Misprint in Line 97: "2th" should be "2nd"

         2) Have been corrected in the manuscript.

  1. I guess the paragraph from line 112 to line 115 is the hypothesis of the study. Suggest to reorganize the language.

          3) In the Microsoft Word template, for the Children journal, it is written in the end of the Introduction: “…Finally, briefly mention the main aim of the work and highlight the principal conclusions…”. That is why we wrote the few lines you are asking about. I would not call it a hypothesis.

  1. In the section "3.4 Leisure-time sport". The parent/guardian reported the number of times leisure-time sport participation, however "hours" are used when you analyzed the data. Please clarify it.

            4) Thank you very much for pointing this out. It's a big mistake that has crept in during the writing process of the manuscript. Parents have been asked about the number of times their child has been to sports in the past week, and therefore it is "number of times per week" and not "number of hours per week. It is now fixed.

Thank you again for using time on our manuscript.

Round 2

Reviewer 1 Report

Thank you for addressing my comments and suggestions. The only things I have suggestions on now are some editing.

Lines 90-101 needs some editing for English, in my opinion. I had to read through it three or four times to fully understand.

Line 354 needs statistics reported in results.
